# Adventitious Shoot Regeneration from Leaf Explants in *Sinningia Hybrida* ‘Isa’s Murmur’

**DOI:** 10.3390/plants11091232

**Published:** 2022-05-02

**Authors:** Honglin Yang, Yihua Yang, Qiang Wang, Jinyu He, Liyun Liang, Hui Qiu, Yue Wang, Lijuan Zou

**Affiliations:** 1Ecological Security and Protection Key Laboratory of Sichuan Province, Mianyang Normal University, Mianyang 621000, China; yanghonglin926@gmail.com (H.Y.); wq18892800791@126.com (Q.W.); lly_liyun@outlook.com (L.L.); qiu2960826082@163.com (H.Q.); 18383094560@163.com (Y.W.); 2State Key Laboratory of Crop Gene Exploration and Utilization in Southwest, Sichuan Agricultural University at Wenjiang, Chengdu 611130, China; 18380444602@163.com; 3College of Life Sciences, China West Normal University, Nanchong 637009, China; jjyyhe26@163.com

**Keywords:** *Sinningia hybrida* ‘Isa’s Murmur’, leaf explant, adventitious shoot regeneration, plant growth regulators

## Abstract

As a valuable ornamental plant, *Sinningia hybrida* ‘Isa’s Murmur’ (*S. hybrida*) has genetic flower diversity, which has great potential to develop different flower characters in the horticultural market. The present study focuses on establishing a practical approach for the sustainable propagation of *S. hybrida*. Compared with aseptic seeding leaves explants, field-grown leaves explants are more suitable for adventitious shoot regeneration. Adding 0.1 mg L^−1^ NAA and 2.0 mg L^−1^ TDZ could obtain the highest adventitious shoot proliferation coefficient (24.5), and the induction rate was 91.7%. The shoot proliferation coefficient (20.7) and the greatest shoot length and induction rate (95.3%) were achieved in 0.1 mg L^−1^ NAA and 2.0 mg L^−1^ BA medium, accompanied by rooting formation. Adding 0.5 mg L^−1^ GA_3_, 1.0 mg L^−1^ BA, and 0.2 mg L^−1^ IBA to MS medium can effectively prolong the regenerated buds for rooting. The best for rooting was 1/2 MS medium containing 0.3 mg L^−1^ IBA, with the maximum number of roots (13.4 per shoot) and survival rate for transplanting (100%). This work aims to build an efficient, definitive, and scalable protocol for *S. hybrida* regeneration useful for large-scale cultivation and even more protoplast fusion and genetic transformation to develop more colorful or fragrant flowers.

## 1. Introduction

The genus *Sinningia* (Gesneriaceae, tribe Sinningieae) is a herbaceous plant with tubers that mainly growth in central and south America. This genus consists of about 70 species with considerable morphological diversity such as different-sized flowers, a wide range of floral color for cultivation, and frequent variations in floral shape and corolla patterning [1,2,3]. Thus, in addition to its high ornamental and economic value, this genus is also an emerging model plant for flower-associated traits research [3]. In general, according to the plant size, leaf size, and flower type, the various species of *Sinningia* can be divided into macro-*Sinningia*, medium *Sinningia*, mini *Sinningia,* and so on [4]. Mini *Sinningia* mainly includes *S. pusilla*, *S. concinna,* and the recently discovered mini lines *S. sellovii* and *S. muscicola* formed by natural or artificial hybridizations [5]. *Sinningia hybrida* ‘Isa’s Murmur’ (*S. hybrida*) belongs to mini *Sinningia* and is an ornamental plant native to southeastern Brazil. They have pendulous, bilaterally symmetrical bell-shaped corollaries in white, pink, or purple [6,7]. Hence, the intragenic floral diversity may also constitute a great potential to develop diverse flower traits in the horticultural market. However, the seed yield of *S. hybrida* is very small, and the seed and vegetative reproduction also have some restraints [8]. Thus, it is necessary to develop an efficient and stable multiplication approach to protect the species and utilize it for horticultural purposes.

In this genus, *S. speciosa* tissue culture has been broadly investigated. How auxin and cytokinin affect shoot regeneration employing leaves, stem segments, or shoot apex as explants has been investigated [9,10,11,12,13,14,15]. The success of *S. speciosa* tissue culture has a specific reference value for constructing the regeneration system of *S. hybrida*. Micropropagation protocols exist for some *Sinningia* species, but there is no literature about the regeneration of *S. hybrida*. This investigation is aimed at establishing an efficient and high-frequency in vitro plant regeneration system of *S. hybrida* from leaf explants. It will facilitate mass multiplication and the use of biotechnology approaches. It provides a certain theoretical basis for protoplast fusion and genetic transformation, for developing more colorful or aromatic *S. hybrida* flowers.

## 2. Results

### 2.1. Effect of Cytokinins on Adventitious Shoot Regeneration

For screening, four cytokinins employed at different concentrations (1.0, 2.0, 3.0 mg L^−1^) were used to induce adventitious shoot regeneration from field-grown leaves. After four weeks of culture, all cytokinin treatments resulted in the adventitious shoot regeneration through callus. Induced response (including differentiation rate, the number of shoots, and plant growth status) on field-grown leaves explant of *S. hybrida* varied with different types of cytokinin (Table 1). The 6-benzylamine (BA) or thidiazuron (TDZ) induction responses were superior to those of zeatin (ZT) and kinetin (Kin). In the medium supplemented with TDZ or BA, a cluster of fleshy and robust shoots with dark-green leaves were induced around the fully callus-leaf (Figure 1A,D), with a shoot differentiation rate of above 70%. In addition, the highest number of adventitious shoots (17.5/explant) was yielded on 2.0 mg L^−1^ TDZ medium. When the medium contained ZT or Kin, the callus and the shoot induction responses were relatively weak. The quantity of shoot produced per explant ranged between 2.7 and 8.5, depending on the cytokinin concentration. Meanwhile, accompanied by the adventitious shoots were slender (in ZT medium) or hyperhydric (in Kin medium) (Figure 1B,C). Therefore, BA and TDZ were used for shoot proliferation in subsequent experiments.

### 2.2. Effect of Plant Growth Regulators on Shoot Proliferation from Two Types of Leaves Explants

In general, combinations of cytokinins and auxin were found to be more effective for shoot regeneration. To obtain the better results, we added different concentrations of NAA to BA- or TDZ-containing medium. Next, we tested the induction response of field-grown leaves (Figure 2A) or aseptic seeding leaves (Figure 2C) explants on MS (Murashige and Skoog) medium containing TDZ or BA with NAA, separately (Table 2). Initially, after 2 weeks of culture, light-green and compact callus was produced on the cut surface of swollen brown leaves (Figure 2B). After being cultured on the same medium for 3 weeks, callus and numerous acervate adventitious shoots were produced from the edges of leaves (Figure 2C). After 4 weeks of culture, well-developed shoots were healthy, featuring dark-green leaves (Figure 2D). For multiplication culture, the multiple shoot clusters were classified into numerous smaller clusters (5–10 buds/clump). Two subcultures acquired numerous cluster buds (Figure 2E,F). Interestingly, the aseptic seeding leaves explants became brown but exhibited no necrosis and continuous high-regeneration ability (Figure 2G). Brown aseptic seeding leaf explants produced a little callus and developed into adventitious shoots after three weeks (Figure 2G) and four weeks (Figure 2H) of culture. On NAA combination by BA medium, prolonging the culture time, healthy prolific shoots were induced and accompanied by numerous adventitious roots (Figure 2I red arrow), but not the TDZ-containing medium.

We found that both field-grown leaves (WL) or aseptic seeding leaves (ASL) browned easily, the induction response of WL explants was better than that of ASL explants, and WL browning rate was significantly lower than that of ASL. The addition of NAA enhanced shoot differentiation rate and proliferation coefficient. The higher concentrations of NAA (0.3 mg L^−1^) or cytokinins (3.0 mg L^−1^) did not obviously promote adventitious shoot regeneration. Shoot length was significantly more increased on the medium of the combination of NAA and BA (mean 1.3 cm) than that of NAA and TDZ (mean 0.7 cm). In general, field-grown leaves explants were more suitable for adventitious shoot regeneration. The combination of TDZ and NAA is more favorable for shoot proliferation. The highest shoot proliferation coefficient (24.5) was reached in 0.1 mg L^−1^ NAA and 2.0 mg L^−1^ TDZ with a 91.7% differentiation rate and dwarfed shoot, whereas the integration of NAA and BA is more suitable for rapid reproduction because it saves the time for shoot elongation and rooting; the shoot proliferation coefficient (20.7) and the most significant shoot length were achieved in 0.1 mg L^−1^ NAA and 2.0 mg L^−1^ BA, featuring a 95.3% differentiation rate.

### 2.3. Adventitious Shoot Elongation and Rooting

For prolonging the length of adventitious shoot, dwarf shoots (Figure 3A) were transferred to a medium containing GA_3_. In all treatments (Table 3), the longest shoot was observed in the medium supplemented with 0.5 mg L^−1^ GA_3_, 0.2 mg L^−1^ IBA, 1.0 mg L^−1^ BA, and 0.1 g L^−1^ casein after three weeks of culture. Regenerated buds (0.5 to 1 cm long, Figure 3A) effectively elongated into robust buds (about 3.0 cm, Figure 3B,C). Furthermore, with the extension of subculture time to two weeks, the quantity of well-developed buds increased obviously (data not shown, Figure 3D). For microshoot rooting, single shoots separated from multiple shoots of explants were cultured on half-strength MS medium supplemented with 0, 0.1, 0.3, and 0.5 mg L^−1^ NAA or IBA. These shoots rooted spontaneously after two weeks (Figure 3E–G). The root numbers per shoot ranged from 7.2 to 13.4, except in control. Maximal root number (13.4 ± 0.4) and length (3.6 ± 0.2 cm) per shoot were discovered in the 0.3 mg L^−1^ IBA-containing cultures, and maximal rooting rate was up to 100% (Table 4).

### 2.4. Acclimatization and Transplantation

The plantlets with well-developed roots were transferred into soil and acclimatized in the culture room. Following one week of hardening, the plants were transferred to a natural field with 100% survival (Figure 4A). Growth states were recorded for three weeks (Figure 4B), four weeks (Figure 4C), and seven weeks (Figure 4D). The emergence of new leaves and flowers of in vitro rooting plants exhibited the success of adaptability.

## 3. Discussion

### 3.1. Effect of PGRs

Four cytokinins successfully induced shoot regeneration. When the content of cytokinin exceeded the optimal level, the induction response decreased. BA is the most effective of the four cytokinins applied in *S. hybrida*. Kuo and Xu reported similar outcomes in species *Sinningia speciosa* [13,16]. TDZ is considered as optimal synthetic cytokinin for the regeneration of many plant systems [17,18,19]. In our study, abundant and dwarf shoots were induced in MS medium containing TDZ, but, even prolonging the culture time, the differentiated adventitious buds failed to elongate (Figure 2F). This discovery might result from lacking degradation of TDZ through cytokinin oxidase enzymes in plant tissues [20,21]. TDZ may inhibit shoot elongation by reducing the endogenous GA concentration [22]. BA or TDZ meet our different demands. On the one hand, the medium containing TDZ could induce more adventitious shoots than BA, but these shoots must perform prolonging before rooting culture. On the other hand, the medium containing BA can directly proliferate and root, thus shortening the culture time. We also found that the hyperhydration regenerated shoots produced by continuous Kin, having negative impacts on the development and regeneration potential of the culture. It is reported that hyperhydricity is related to cytokinin concentration and type [23,24,25]. In the present study, Kin causes shoot hyperhydration, this result aligning with our previous discoveries on *Ajuga lupulina* Maxim [26].

Auxin alone or in combination with cytokinin was exhibited to stimulate shoot generation of plants such as *Sinningia speciosa* [16] and *Saintpaulia* [27,28,29]. In our experiments, the addition of NAA significantly improved the number and proliferation coefficient of adventitious shoots. More importantly, in the present study, BA is the most optimal cytokinin. On the medium of NAA and BA, adventitious shoots were produced, accompanied by the adventitious root formation. This would shorten the culture period and save costs. This conclusion has been reported in *Salvia plebeia* species [30]. Cheesman and Aremu also proved the important role of various plant hormones, comprising NAA, in promoting adventitious bud generation [31,32]. The ratio of cytokinin to auxin holds the key to shoot regeneration and proliferation. In our study, the combinations of 0.1 mg L^−1^ auxin and 2.0 mg L^−1^ cytokinin were more beneficial to shoot proliferation; above the optimal level, bud formation rate decreased. Cytokinin dose affects bud proliferation [33]. Our research indicated that longer exposure duration and higher concentrations of TDZ were toxic to explants, causing browning. These results are along the same lines for *Jatropha curcas* L. [34].

### 3.2. Leaf as Explant Source for Regeneration

As a convenient and abundant raw material, leaves were used in the investigation since they usually exhibit better regeneration potential than explants from mature tissues [35]. Moreover, in vitro leaf explants are more suitable for genetic transformation research as they are sterile, and gene stacking technology is feasible [16,36]. Many studies have successfully established the tissue culture system using leaves as explants, such as *Sinningia speciosa* [16] and *Saintpaulia* species [27,28,29]. In our study, both field-grown leaves and aseptic seeding leaves as explants successfully induced shoot regeneration from *S. hybrida* by organogenesis pathway. Although the aseptic seeding leaves brown easily, they still sustained high differentiation capacity. The phenomenon has been reported in species of *Ajuga lupulina* [26]. We suspect that light intensity or light quality may cause browning, but this is not enough to adversely affect the differentiation ability of leaf explants. In addition, some studies have reported how light intensity and light quality affect shoot regeneration capacity [37]. Dark-green and thick leaves, such as field-grown leaves, have strong photosynthesis ability. In contrast, light-green and thin aseptic seeding leaves have weak photosynthesis, so the induction response of field-grown leaves is superior to that of aseptic seeding leaves of *S. hybrida*.

### 3.3. Elongation, Rooting

In our study, although TDZ has been proven to be a powerful shoot organogenic agent to induce the proliferation of *S. hybrida*, the adverse impacts of TDZ on explants and shoots were reported in tissue culture, comprising inhibition of bud elongation [38,39,40]. Therefore, the regenerated shoot must undergo elongated growth before it can be used for rooting. GA_3_ was effectively improving shoot elongation [41].

The rooting rate and root number elevated as auxin (IBA or NAA) concentration increased. When the optimum content was exceeded, induction rate decreased. IBA is more suitable for rooting of *S. hybrida*, which is consistent with the findings in other reports [42,43,44,45].

## 4. Materials and Methods

### 4.1. Plant Material, Basal Medium, and Culture Conditions

*Sinningia hybrida* ‘Isa’s Murmur’ was purchased from Horticulture Co., Ltd. in Sichuan. The plant was certified by Professor Minghua Luo of Mianyang Normal University. Leaf was excised from mature greenhouse-grown plantlets. Explants were initially rinsed for 30 min employing water, and then surface-sterilized, adopting 75% ethanol, for 15 s before 0.1% (*w*/*v*) HgCl_2_-mediated disinfection for 5 min, then rinsed five times, applying distilled water.

They were placed on Murashige and Skoog (MS) basal medium [46] with different plant growth regulators (PGRs) for shoot induction. All culture medium contained 3% (*w*/*v*) sucrose and 0.7% agar. The pH of media was adjusted to 5.6 with 0.1 mol L^−1^ NaOH before autoclaving for 15 min at 121 °C. All PGRs (thidiazuron (TDZ), zeatin (ZT), kinetin (Kin), 6-benzylaminopurine (BA), 3-Indole-butyric acid (IBA), naphthalene acetic acid (NAA), and Gibberellin A_3_ (GA_3_) were purchased from Sigma-Aldrich (St. Louis, MI, USA). The cylindrical culture bottle had a diameter of 7.0 cm and a height of 8.0 cm. All explants and plantlets were cultured at 25 ± 1 °C under a 12 h light cycle with a light intensity of 30 µmol m^−2^ s^−1^. At the same time, we selected the different length of experiments according to the development time of adventitious shoots, which is different to that of adventitious roots.

### 4.2. Screening for Cytokinins

To study how various cytokinins affected adventitious bud induction, the screening cytokinins contained 1.0, 1.5, or 2.0 mg L^−1^ of TDZ, ZT, Kin, or BA (Table 1). Field-grown leaf explants were excised from a healthy mother plant. Sterilized leaves were minced to 0.5 × 0.5 cm^2^ followed by medium with different cytokinins, with their adaxial surface facing down. After four weeks of culture, the mean shoot quantity per explant was recorded.

### 4.3. Plant Growth Regulators (PGRs) on Adventitious Bud Proliferation

Apical bud explants were utilized for primary culture on medium with IBA 0.2 mg L^−1^ and BA 0.5 mg L^−1^ to obtain aseptic seeding leaves. Since the ratio of cytokinin to auxin determines the induction of adventitious buds, three different cytokinin concentrations were selected. The combination effects of different concentrations of NAA (0.1, 0.3 mg L^−1^) with BA or TDZ (1.0, 2.0, 3.0 mg L^−1^) were investigated. Field-grown and aseptic seeding leaves were used for explants (Table 2). After incubation for 4 weeks, the adventitious bud proliferation coefficient was performed below: quantity of buds after inoculation/quantity of buds before inoculation.

### 4.4. Adventitious Shoot Elongation, Rooting, Acclimatization and Transplantation

Adventitious bud clusters were carefully separated from dwarf shoots and some intact basal parts were retained. Medium containing 0.2 mg L^−1^ NAA or IBA, 0.5 or 1.0 mg L^−1^ BA, 0.5 mg L^−1^ GA_3_, and 0.1 g L^−1^ casein hydrolysate was used for shoot elongation assay (Table 3). The average length of shoots was measured after three weeks.

Reducing the levels of inorganic salt and sucrose contribute to increase root induction. In order to regenerate plantlets from robust shoots, shoots from half-strength MS were supplemented with 0.1, 0.3, and 0.5 mg L^−1^ NAA or IBA for rooting. The half-strength MS contained half-strength MS salts and 1% (*w*/*v*) sucrose. After two weeks of culture, the rooting rate and root length were measured. The developed roots were transferred into plastic pots containing nutrient soil (80% sandy loam, 10% vermiculite, and 10% perlite). The greenhouse temperature was maintained at 25 ± 1 °C, the light intensity was 30 µmol m^−2^ s^−1^, and the relative humidity was 75 to 85%.

### 4.5. Statistical Analysis

All the experiments were repeated three times (thirty explants per treatment). All data were analyzed by one-way analysis of variation (ANOVA) using SPSS version 18.0 for Windows (Chicago, IL, USA). The mean with standard error (±SE) was presented. Significantly different treatments were tested by a Duncan’s multiple comparison test (*p* = 0.05).

## 5. Conclusions

In this investigation, the effect of PGRs and explant-type on callus induced and differentiation, shoot proliferation, elongation, and rooting of *Sinningia hybrida* ‘Isa’s Murmur’ were performed. The results revealed that field-grown leaves are superior to aseptic seeding leaves for micropropagation. In addition, we found that BA and TDZ played a vital role in shoot proliferation. A large number of adventitious shoots were produced on the medium containing NAA and BA, accompanied by mass adventitious roots forming, and these rooting plants were successfully transplanted. This kind of proliferation and rooting culture can be completed on the same medium, which not only shortened the cultivation time but also saved costs. Rooting culture for TDZ-induced adventitious shoots, IBA, is more suitable for rooting (rooting rate 100%) than NAA (rooting rate 90.8%) of *S. hybrida.* Overall, we set up a high-frequency regeneration system for leaves explants of *S. hybrida.* It helps conservation of germplasm resources, rapid propagation, and genetic transformation for the genus.

## Figures and Tables

**Figure 1 plants-11-01232-f001:**
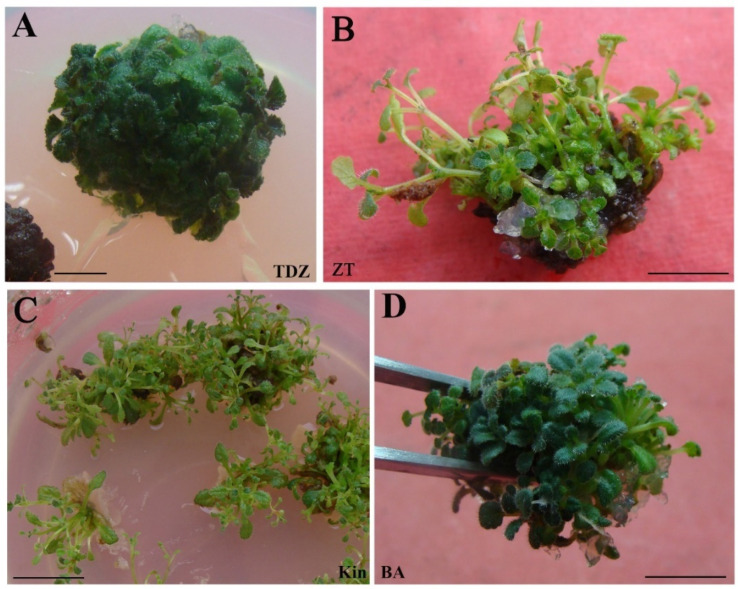
Impact of cytokinins on direct adventitious bud regeneration of *Sinningia hybrida* ‘Isa’s Murmur’. Adventitious shoot formation on 2.0 mg L^−1^ TDZ (**A**), ZT (**B**), Kin (**C**), and BA (**D**) medium after four weeks of culture. *Bar* = 5 mm (**A**), 2 cm (**B**), 1 cm (**C**,**D**).

**Figure 2 plants-11-01232-f002:**
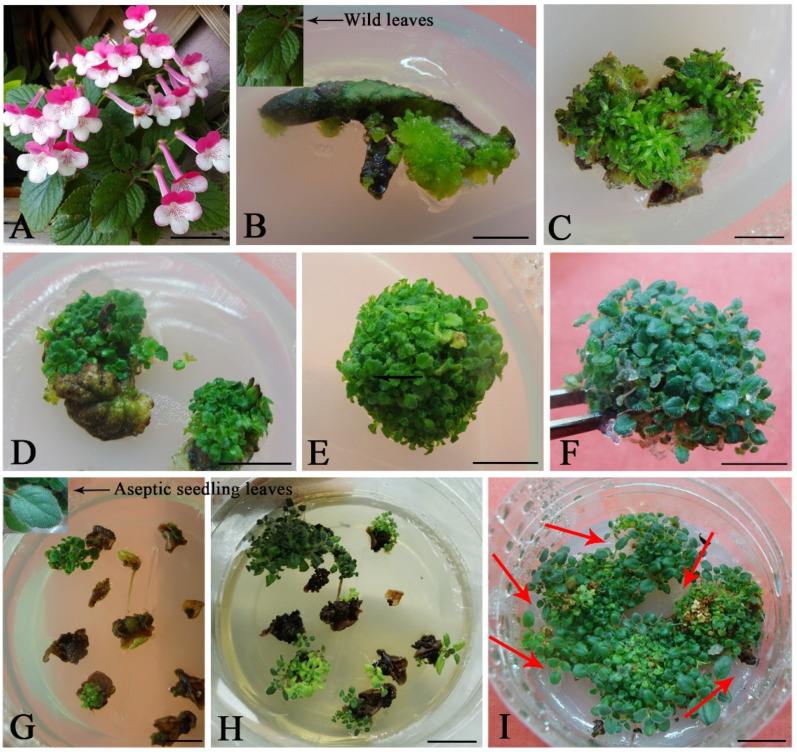
Direct adventitious shoot regeneration and proliferation from *Sinningia hybrida* ‘Isa’s Murmur’ leaf explants on MS medium supplemented with 2.0 mg·L^−1^ BA and 0.1 mg·L^−1^ NAA. (**A**) Widely grown *S. hybrida* plants. (**B**–**F**) Adventitious shoot regeneration from field-grown *S. hybrida* leaves explants. (**B**) Acervate shoots arise from field-grown leaves after a 2-week of culture. (**C**) Formation of adventitious shoots on the leaf explant surface following culture for three weeks. (**D**) Prolific adventitious shoots following four weeks of culture. (**E**,**F**) Proliferation culture. (**G**–**I**) Adventitious shoot regeneration from aseptic seeding leaves explants on MS medium supplemented with 2.0 mg·L^−1^ BA and 0.3 mg·L^−1^ NAA. (**G**) Direct adventitious shoot generation from aseptic seeding leaves following three weeks of culture. (**H**) Proliferation adventitious shoots following four weeks of culture. (**I**) After 6 weeks, prolific adventitious buds were produced from the bottom of the bud with abundant adventitious roots (red arrow indicates). *Bar* = 2 cm (**A**), 5 mm (**B**–**D**), 1 cm (**E**–**I**).

**Figure 3 plants-11-01232-f003:**
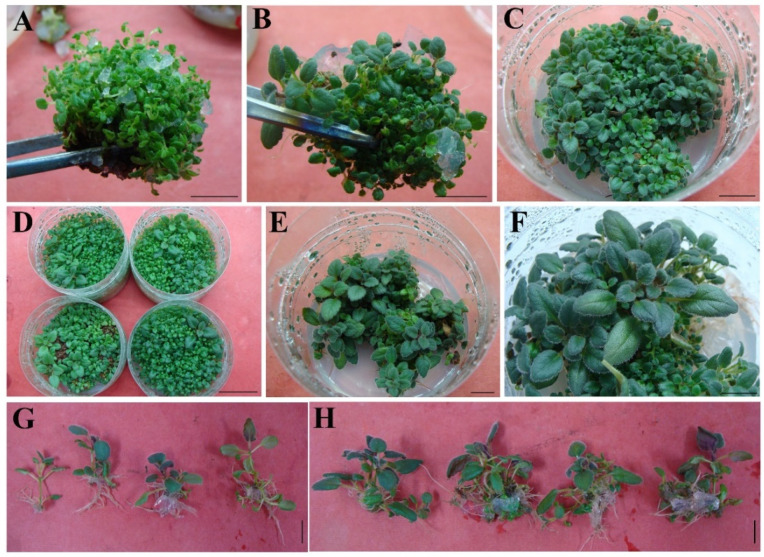
Regenerated shoot elongation and rooting of *Sinningia hybrida* ‘Isa’s Murmur’. (**A**) Adventitious buds on MS medium containing 2.0 mg L^−1^ TDZ and 0.1 mg L^−1^ NAA. (**B**–**D**) Elongated buds from MS medium containing 0.5 mg L^−1^ GA_3_ combined with 0.1 g L^−1^ casein, 0.2 mg L^−1^ IBA, and 1.0 mg L^−1^ BA. (**E**,**F**) Rooting of buds on half-strength MS medium featuring 0.5 mg L^−1^ IBA. (**G**) Rooting of a plantlet for 2 weeks. (**H**) Rooting of shoots on the medium of the combination of NAA and BA. *Bar* = 1 cm (**A**–**H**), 3 cm (**D**).

**Figure 4 plants-11-01232-f004:**
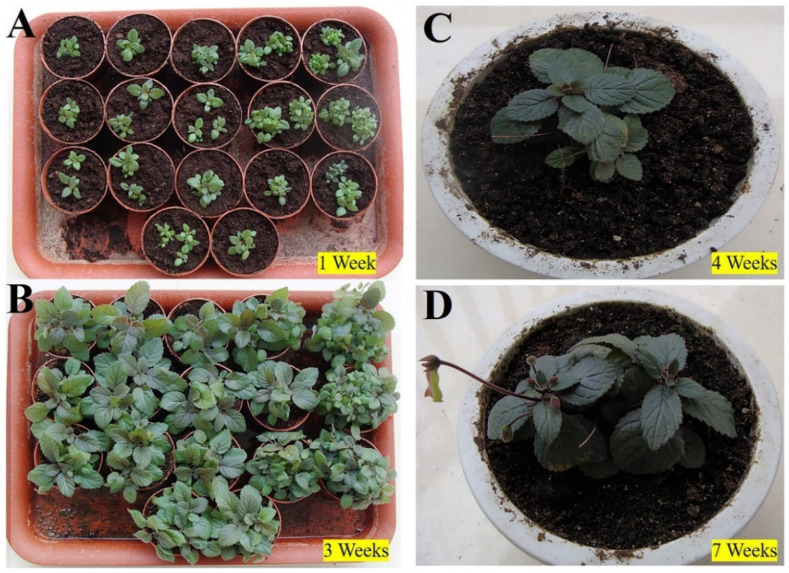
Acclimatization. Acclimatized plants by rooting in vitro after one week (**A**), three weeks (**B**), four weeks (**C**), and seven weeks (**D**), respectively.

**Table 1 plants-11-01232-t001:** Effect of cytokinins on direct adventitious shoot regeneration of *Sinningia hybrida* ‘Isa’s Murmur’.

Cytokinin (mg·L^−^^1^)	Differentiation Rate (%)	No. of Shoot (Per Explant)	Visible Appearance	
			The Color of the Leaves	Shoot Growth State
TDZ 1.0	63.0 ± 2.9 b	12.5 ± 1.2 bc	Dark-green	Healthy
TDZ 2.0	74.0 ± 2.0 a	17.5 ± 0.6 a	Dark-green	Healthy
TDZ 3.0	61.9 ± 3.6 b	14.3 ± 0.8 ab	Dark-green	Healthy
ZT 1.0	43.7 ± 4.3 d	5.8 ± 0.4 cd	Light-green	Hyperhydricity
ZT 2.0	54.6 ± 1.8 bc	8.5 ± 0.9 c	Light-green	Hyperhydricity
ZT 3.0	51.4 ± 2.8 bc	6.0 ± 0.4 cd	Light-green	Hyperhydricity
Kin 1.0	34.4 ± 3.0 e	2.7 ± 0.7 e	Light-green	Hyperhydricity
Kin 2.0	42.1 ± 2.7 d	6.3 ± 0.8 cd	Light-green	Hyperhydricity
Kin 3.0	40.9 ± 2.9 d	4.1 ± 0.4 d	Light-green	Hyperhydricity
BA 1.0	68.7 ± 3.4 ab	9.5 ± 0.9 ab	Dark-green	Healthy
BA 2.0	72.7 ± 2.7 a	14.9 ± 0.9 ab	Dark-green	Healthy
BA 3.0	66.9 ± 2.2 ab	13.9 ± 1.3 ab	Dark-green	Healthy

Results were recorded after four weeks of culture. Mean ± SD followed by same letters within a column are not significantly different (*p* < 0.05).

**Table 2 plants-11-01232-t002:** Effect of plant growth regulators on shoot proliferation from leaf explants o *Sinningia hybrida* ‘Isa’s Murmur’.

Explant	NAA (mg·L^−1^)	TDZ (mg·L^−1^)	BA (mg·L^−1^)	Browning Rate (%)	Differentiation Rate (%)	Shoot Length (cm)	Shoot Proliferation Coefficient
WL	0.1	1.0	-	71.3 ± 3.0 cd	86.0 ± 3.4 bc	0.4 ± 0.1 b	16.5 ± 1.0 bc
WL	0.1	2.0	-	67.8 ± 2.6 de	91.7 ± 2.0 ab	0.7 ± 0.2 b	24.5 ± 2.0 a
WL	0.1	3.0	-	72.9 ± 2.2 cd	88.4 ± 1.3 bc	0.5 ± 0.2 b	18.6 ± 1.1 b
WL	0.3	1.0	-	73.6 ± 1.9 cd	78.3 ± 1.5 cd	0.8 ± 0.2 b	12.5 ± 0.9 de
WL	0.3	2.0	-	77.2 ± 2.8 c	85.5 ± 3.0 b	0.9 ± 0.4 b	16.6 ± 1.3 bc
WL	0.3	3.0	-	82.6 ± 2.9 b	81.6 ± 1.4 c	0.8 ± 0.1 ab	11.6 ± 1.2 de
WL	0.1	-	1.0	70.5 ± 4.6 cd	88.4 ± 2.3 b	0.9 ± 0.2 b	19.4 ± 1.3 b
WL	0.1	-	2.0	66.0 ± 2.4 de	95.3 ± 1.7 a	1.3 ± 0.2 a	20.7 ± 1.5 b
WL	0.1	-	3.0	80.2 ± 2.8 b	86.1 ± 1.6 bc	1.1 ± 0.2 a	18.1 ± 0.8 b
WL	0.3	-	1.0	74.8 ± 3.1 cd	84.3 ± 3.9 bc	1.0 ± 0.3 ab	9.6 ± 1.9 e
WL	0.3	-	2.0	60.9 ± 2.4 de	84.6 ± 4.2 bc	1.1 ± 0.3 a	13.0 ± 2.6 de
WL	0.3	-	3.0	80.5 ± 2.3 bc	82.6 ± 2.6 bc	1.0 ± 0.3 ab	16.0 ± 2.0 bc
ASL	0.1	1.0	-	80.7 ± 1.5 bc	55.6 ± 3.3 f	0.6 ± 0.2 b	10.0 ± 1.4 e
ASL	0.1	2.0	-	83.9 ± 3.2 b	67.3 ± 3.6 e	0.9 ± 0.3 ab	12.6 ± 2.1 de
ASL	0.1	3.0	-	84.5 ± 3.7 b	64.2 ± 2.7 e	0.6 ± 0.1 b	11.0 ± 2.3 de
ASL	0.3	1.0	-	84.5 ± 3.9 b	47.3 ± 1.7 g	0.6 ± 0.2 b	16.3 ± 0.8 bc
ASL	0.3	2.0	-	81.6 ± 1.6 bc	56.8 ± 3.4 f	0.8 ± 0.2 b	21.5 ± 1.4 ab
ASL	0.3	3.0	-	94.3 ± 1.0 a	40.2 ± 3.4 h	0.7± 0.1 b	17.7 ± 1.8 b
ASL	0.1	-	1.0	86.2 ± 1.8 b	57.9 ± 1.9 f	0.5 ± 0.2 b	12.2 ± 1.9 de
ASL	0.1	-	2.0	88.2 ± 2.4 b	69.7 ± 3.4 e	0.8 ± 0.3 b	14.7 ± 1.1 d
ASL	0.1	-	3.0	89.4 ± 2.3 b	57.5 ± 3.2 f	0.6 ± 0.3 b	9.1 ± 2.1 e
ASL	0.3	-	1.0	92.6 ± 2.1 a	53.7 ± 4.0 f	0.5 ± 0.2 b	16.2 ± 1.5 bc
ASL	0.3	-	2.0	83.7 ± 3.5 b	62.9 ± 2.6 e	0.7± 0.2 b	20.0 ± 1.3 b
ASL	0.3	-	3.0	95.9 ± 1.6 a	60.6 ± 1.3 e	0.7 ± 0.2 b	18.0 ± 0.9 b

Results were recorded after four weeks of culture. Mean ± SD followed by same letters within a column are not significantly different (*p* < 0.05). Field-grown leaves (WL); aseptic seeding leaves (ASL).

**Table 3 plants-11-01232-t003:** Effect of plant growth regulators on shoot elongation of *Sinningia hybrida* ‘Isa’s Murmur’.

PGRs for Shoot Elongation (mg·L^−1^)	Shoot Length (cm)	Observed Results
NAA0.2 + BA0.5 + GA_3_0.5 + 100.0 casein	1.4 ± 0.2 c	Light-green; soft and fragile; dwarf shoot
NAA0.2 + BA1.0 + GA_3_0.5 + 100.0 casein	2.2 ± 0.5 b	Light-green; soft and fragile; dwarf shoot
IBA0.2 + BA0.5 + GA_3_0.5 + 100.0 casein	2.7 ± 0.4 a	Dark-green; robust shoot
IBA0.2 + BA1.0 + GA_3_0.5 + 100.0 casein	3.0 ± 0.3 a	Dark-green; robust shoot

Results were recorded after three weeks of culture. Mean ± SD followed by same letters within a column are not significantly different (*p* < 0.05).

**Table 4 plants-11-01232-t004:** Effects of auxins on root induction of *Sinningia hybrida* ‘Isa’s Murmur’.

Treatment (mg·L^−1^)	Culture Time		Rooting Rate (%)	Mean Root Number	Root Length (cm)
	2 Weeks	4 Weeks			
0	−	−	38.0 ± 3.2 e	1.8 ± 1.4 e	1.3 ± 0.3 e
IBA 0.1	−	+	87.6 ± 0.5 bc	9.5 ± 0.2 c	2.5 ± 0.1 cd
IBA 0.3	+	+	100 ± 0.1 a	13.4 ± 0.4 a	3.6 ± 0.2 a
IBA 0.5	+	+	99.1 ± 0.8 a	11.4 ± 0.7 b	2.8 ± 0.4 bc
NAA 0.1	−	−	75.5 ± 0.6 d	7.2 ± 0.5 d	2.1 ± 0.3 d
NAA 0.3	−	+	90.8 ± 0.4 b	9.4 ± 0.5 c	3.3 ± 0.5 a
NAA 0.5	−	+	81.6 ± 1.6 cd	8.0 ± 0.2 dc	2.8 ± 0.1 bc

Results were recorded after two weeks of culture. Mean ± SD followed by same letters within a column are not significantly different (*p* < 0.05). All treatments included half-strength MS medium. + indicates root formation, and − indicates that roots were not formed yet within the observed time period.

## Data Availability

All data generated or analyzed during this study are included in this published article.

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
