# Peer review of "Adventitious Shoot Regeneration from Leaf Explants in *Sinningia Hybrida* ‘Isa’s Murmur’"

_plants, 2022, doi:10.3390/plants11091232_

Round 1

Reviewer 1 Report

The manuscript presents a set of experiments on shoot regeneration in Sinningia hybrida which could be important for horticultural industry. Unfortunately, in many parts of the MS there are discrepancies, editorial mistakes (esp. lacking space bars and full stops) and linguistic faults which should be definitely improved.

Some of the faults and suggestions are stated below.

Introduction

Please, rewrite the sentence in order to be written correctly in English and logically: “Besides their important ornamental and economic value, making it an emerging model plant for flower traits research”.

„S.sellovii and S.Muscicola” should be “S. sellovii and S. muscicola”  - appropriate space bars and small letter in “muscicola”.

“… various species of Sinningia can be divided into macro-Sinningia, medium Sinningia, mini Sinningia …” To which group Sinningia hybrida ‘Isa’s Murmur’ (S. hybrida) belongs?

“Thus, it is urgent to develop …” – the word “urgent” seems strange in this context. Please, rewrite.

“In this genus, S. speciosa …” - lack of italics for S. speciose. Please, remember to write italics for Latin names through the whole text of the MS.

Some specific knowledge concerning plant growth regulators could be a great advantage in this section.

Results

How the doses of growth regulators were chosen to this experiment?

Why for some experiments MS medium and for the others half-strength MS medium were used?

Why the length of experiments is different?

2.1.

Abbreviations are not explained when first mentioned in the text, e.g., TDZ, ZT, BA, Kin. All these abbreviations are explained in 4.2. Please, correct it.

What “KT medium” means? It is not explained.

Table 1. What is the exact definition of “Shoot growth state”? And ““+” indicates shoot growth condition, and the more +, the better growth condition” also needs explanation or rewriting. It is unclear what was evaluated.

2.2

“Next, we tested the induction response of wild or aseptic seeding leaves explants on MS medium containing TDZ and BA individually or with NAA, separately” or better “We tested the induction response of wild or aseptic seeding leaves explants on MS medium containing TDZ or BA with NAA”?

Again abbreviations without explanation, e.g., MS medium.

Fig. 2A – not mentioned in the text.

“…no necrosis, continuous …” or better „ … no necrosis and continuous…”?

“a few callus”?

“… prolific shoots and mass adventitious roots on NAA combination by BA medium …” – please, clarify English and level of details (do you mean higher mass?).

“induction response” and ”induction rate” need clarification and explanation

“didn’t”???

Please, check “was” and “were” in the whole text and improve English where necessary.

“bud elongation and rooting, the bud proliferation coefficient” are mentioned in the text but I cannot see such parameters in the tables. Where are they?

Why BA and TDZ in Tab. 2 are in other concentrations than in Tab. 1?

“(B) acervate …” should be “(B) Acervate …” (capital letter).

“(C) Formation of really adventitious …” or “(C) Formation of adventitious …” (without really)?

2.3.

“For prolonging the length of adventitious shoot, dwarf shoots (Figure 3A) were transferred to a medium containing GA3 to prolong growth” should be better “For prolonging the length of adventitious shoot, dwarf shoots (Figure 3A) were transferred to a medium containing GA3” because “to prolong growth” is the repetition.

“except in controls” or „except in control”??

“rooting frequency” – needs clarification and explanation.

Why data from Tab. 3 were achieved after 21 days but in Tab. 4 after 2 and 4 weeks?

2.4.

First and second sentences need English correction.

3.1.

“But even prolong …” or „But even prolonging …”?

What do you mean “According to our different needs …”??

“ … aligning with our present discoveries [27]” – Do you mean your discoveries or others? Why are you citing the literature here? It needs correction.

“In our study, compared to cytokinin alone, the additive of auxin NAA, the number of adventitious buds, and the proliferation coefficient were significantly improved” – Unclear, improve English, please.

“On the medium of NAA and BA, adventitious shoots were produced accompanied by mass adventitious roots” – Do you mean higher mass?

“in species Salvia plebeia” or „ in Salvia plebeian species”?

3.2.

“capacity, the phenomenon …” or „capacity. The phenomenon …”??

“in some species Ajuga lupulina” or “in some species of Ajuga lupulina”??

“works of literaturę” or simply „papers”?

3.3.

The first paragraph – What about your results? The same or opposite to the literature cited there?

4.1.

The second paragraph – improve English, please. Again abbreviation was not explained when first used (PGRs).

4.2.

Two sentences need English correction.

“screening medium” needs definition.

4.2., 4.4., 4.5. – lack of clarity in repetitions. Please, rewrite and calrify.

4.3.

IBA – abbreviation not explained.

4.4.

Clarify – what was the content of control?

Please, improve English in the sentence “Transfer plants …”

4.5.

The explanation of LSD is wrong. Please, correct it.

“obviously different” or “different”??

“statistic analysis”? – English

Add company name for software.

“we found …” or “we found that …”?

Unclear sentence, needs English correction – “A large number of adventitious buds were produced on the medium containing NAA and BA, accompanied by adventitious roots, these rooting plants were successfully transplanted directly, which not only shortened the cultivation time but also saved the cost”.

“IBA is more suitable for rooting of S.hybrida” – than what? why?

References – need unification, e.g.,

nr. 9 (In Vitro Cell Dev Biol Plant.) vs. nr. 31 (In Vitro Cell.Dev.Biol.-Plant.);

nrs 23, 27 – title – small letters like in a sentence;

lack of italics in Latin names – nrs 25, 36, 43.

Author Response

Reviewer #1:

Introduction

Question 1. Please, rewrite the sentence in order to be written correctly in English and logically: “Besides their important ornamental and economic value, making it an emerging model plant for flower traits research”.

Answer 1: Thanks for your suggestion, we have been revised.

Question 2. “S.sellovii and S.Muscicola” should be “S. sellovii and S. muscicola” - appropriate space bars and small letter in “muscicola”.

Answer 2: Thanks for your suggestion, we have been revised.

Question 3: “In this genus, S. speciosa …” - lack of italics for S. speciose. Please, remember to write italics for Latin names through the whole text of the MS.

Answer1: We have revised these errors according to your suggestion.

Question 4. “… various species of Sinningia can be divided into macro-Sinningia, medium Sinningia, mini Sinningia …” To which group Sinningia hybrida ‘Isa’s Murmur’ (S. hybrida) belongs?

Answer 4:  Thanks for your suggestion, Sinningia hybrida ‘Isa’s Murmur’ (S. hybrida) belongs mini Sinningia. We have rewritten clearly in the article.

Question 5. “Thus, it is urgent to develop …” – the word “urgent” seems strange in this context. Please, rewrite.

Answer 5: We have been rewritten the word “urgent” and marked it in red.

Results

Question 1. How the doses of growth regulators were chosen to this experiment?

Answer 1: The selection of plant growth regulators is based on literature review previously and preliminary experimental results.

Question 2. Why for some experiments MS medium and for the others half-strength MS medium were used?

Answer 2: Many previous studies have shown that half-strength MS medium is suitable for rooting culture. Reducing the levels of inorganic salt and sucrose contribute to increase root induction has been proved in many herb species. However, MS medium has more comprehensive nutrient composition and is more suitable for shoot induction. These has been well documented.

Question 3. Why the length of experiments is different?

Answer 3: Because in the different growth phases of plants, there are different development times.

2.1

Question 1. Abbreviations are not explained when first mentioned in the text, e.g., TDZ, ZT, BA,Kin. All these abbreviations are explained in 4.2. Please, correct it.

Answer 1: We have been revised.

Question 2. Again abbreviations without explanation, e.g., MS medium.

Answer 2: We have been revised in the text.

Question 3. What “KT medium” means? It is not explained.

Answer 3: “KT medium” has been modified to “Kin medium”.

Question 4. Table 1. What is the exact definition of “Shoot growth state”? And ““+” indicates shoot growth condition, and the more +, the better growth condition” also needs explanation or rewriting. It is unclear what was evaluated.

Answer 4: We have added some detail in table 1

Results 2.2

Question 1. “Next, we tested the induction response of field- grown or aseptic seeding leaves explants on MS medium containing TDZ and BA individually or with NAA, separately” or better “We tested the induction response of field- grown or aseptic seeding leaves explants on MS medium containing TDZ or BA with NAA”?

Answer 1:  We have been revised according to reviewer’s suggestion.

Question 2.  Fig. 2A – not mentioned in the text.

Answer 2:  We have added it in the text.

Question 3. “…no necrosis, continuous …” or better „ … no necrosis and continuous…” ?

Answer 3:  We have been revised according to reviewer’s suggestion.

Question 4. “a few callus”?

Answer 4:  “a few callus” has been modified to “a little callus”

Question 5. “… prolific shoots and mass adventitious roots on NAA combination by BA medium …” – please, clarify English and level of details (do you mean higher mass?).

Answer 5: I don’t get your mean. What we mean is that a large number of adventitious buds and roots are produced.

Question 6. “induction response” and “induction rate” need clarification and explanation

Answer 6: “Induced response” includes induction rate and plant growth status.

 “induction rate” has been modified to “differentiation rate” in the text.

Question 7. “didn’t”???

Answer 7: We have been revised.

Question 8. Please, check “was” and “were” in the whole text and improve English where necessary.

Question 9. “(B) acervate …” should be “(B) Acervate …” (capital letter).

Question 10. “(C) Formation of really adventitious …” or “(C) Formation of adventitious …” (without really)?

Answer 7,8,9,10: Thank the reviewer for these kindly suggestions, we have revised these errors according to your comments and marked them in red.

Question 10. “bud elongation and rooting, the bud proliferation coefficient” are mentioned in the text but I cannot see such parameters in the tables. Where are they?

Answer 10: “bud elongation” and “the bud proliferation coefficient” at table 2, “rooting” at Figure 2I red arrow.

Question 11. Why BA and TDZ in Tab. 2 are in other concentrations than in Tab. 1?

Answer 11: In Tab. 1, we only screened for cytokinins, the optimal concentration is 1.5mg/L. Whereas in table 2 we added auxin NAA. Since the ratio of cytokinin to auxin determines the induction of adventitious buds, we increased the cytokinin concentration.

Results 2.3

Question 1. “For prolonging the length of adventitious shoot, dwarf shoots (Figure 3A) were transferred to a medium containing GA3 to prolong growth” should be better “For prolonging the length of adventitious shoot, dwarf shoots (Figure 3A) were transferred to a medium containing GA3” because “to prolong growth” is the repetition.

Answer 1:  We have been revised according to reviewer’s suggestion.

Question 2. “except in controls” or „except in control”??

Answer 2:  We have been revised according to reviewer’s suggestion. “except in controls” has been modified to “except in control”.

Question 3. “rooting frequency” – needs clarification and explanation.

Answer 3: We have been revised according to reviewer’s suggestion. “rooting frequency” has been modified to “rooting rate”.

Question 4. Why data from Tab. 3 were achieved after 21 days but in Tab. 4 after 2 and 4 weeks?

Answer 4: Because the development time of adventitious shoots is different from that of adventitious roots.

Results 2.4

Question 5. First and second sentences need English correction.

Answer 5: we have revised these sentences and marked them in red.

3.1

Question 1. “But even prolong …” or „But even prolonging …”?

Answer 1: We have been revised according to reviewer’s suggestion.

Question 2. What do you mean “According to our different needs …”??

Answer 2: BA or TDZ meet our different demands. On the one hand, the medium containing TDZ could induce more adventitious shoots than BA, but these shoots must be performed pro-longing and rooting. on the other hand, the medium containing BA can directly proliferate and rooting, thus shortening the culture time. 

Question 3. “ … aligning with our present discoveries [27]” – Do you mean your discoveries or others? Why are you citing the literature here? It needs correction.

Answer 3: We have rewritten the sentence.

Question 4. “In our study, compared to cytokinin alone, the additive of auxin NAA, the number of adventitious buds, and the proliferation coefficient were significantly improved” – Unclear, improve English, please.

Answer 4: We have rewritten the sentence.

Question 5. “On the medium of NAA and BA, adventitious shoots were produced accompanied by mass adventitious roots” – Do you mean higher mass?

Answer 5: No, we wanted to express “shoots were produced accompanied by the formation of roots”. We have been revised.

Question 6. “in species Salvia plebeia” or „ in Salvia plebeian species”?

Answer 6: We have been revised according to reviewer’s suggestion.

3.2

Question 1. “capacity, the phenomenon …” or „capacity. The phenomenon …”??

Question 2. “in some species Ajuga lupulina” or “in some species of Ajuga lupulina”??

Question 3. “works of literaturę” or simply „papers”?

Answer 1,2,3: Thank the reviewer for these kindly suggestions, we have revised according to your comments and marked them in red.

3.3

Question 1. The first paragraph – What about your results? The same or opposite to the literature cited there?

Answer 1:  In order to better express our ideas, we deleted some words.

4.1

Question 1. The second paragraph – improve English, please. Again abbreviation was not explained when first used (PGRs).

Answer 1: We have been revised according to reviewer’s suggestion.

4.2

Question 1. Two sentences need English correction.

Answer 1: We have been revised according to reviewer’s suggestion.    

Question 2. “screening medium” needs definition.

Answer 2: We have been revised, “medium with different cytokinins”.

Question .3. 4.2., 4.4., 4.5. – lack of clarity in repetitions. Please, rewrite and calrify.

Answer 3: We have been revised according to reviewer’s suggestion.

4.3

Question 1. IBA – abbreviation not explained.

Answer 1: we have added according to your comments and marked them in red.

4.4

Question 1. Clarify – what was the content of control?

Answer 1: The control was medium without auxins

Question 2. Please, improve English in the sentence “Transfer plants …”

Answer 2: We have added according to your comments.

4.5

Question 1. The explanation of LSD is wrong. Please, correct it.

Answer 1: We have added according to your comments.

Question 2. “obviously different” or “different”??

Answer 2: We have added according to your comments.

Question 3. “statistic analysis”? – English

Answer 3: We have rewritten the sentence.

Question 4. Add company name for software.

Answer 4: We have added it.

5

Question 1. “we found …” or “we found that …”?

Answer 1: We have been revised according to reviewer’s suggestion.

Question 2. Unclear sentence, needs English correction – “A large number of adventitious buds were produced on the medium containing NAA and BA, accompanied by adventitious roots, these rooting plants were successfully transplanted directly, which not only shortened the cultivation time but also saved the cost”.

Answer 2:  We have been revised.

Question 3. “IBA is more suitable for rooting of S.hybrida” – than what? why?

Answer 3: We have been revised according to reviewer’s suggestion.

References

Question 1. nr. 9 (In Vitro Cell Dev Biol Plant.) vs. nr. 31 (In Vitro Cell.Dev.Biol.-Plant.);

Question 2. nrs 23, 27 – title – small letters like in a sentence;

Question 3. lack of italics in Latin names – nrs 25, 36, 43.

Answer 1,2,3: Thank the reviewer for these kindly suggestions, we have revised these errors according to your comments and marked them in red.

Reviewer 2 Report

Dear authors,

The introduction is too short and could be improved;

I think that Fisher's LSD test was the best choice. This is the most liberal post-hoc test that easily finds significant differences. I recommend to use the more conservative Tukey's test to identify only those significant differences which most likely represent biological meaning.

Author Response

Dear editors:

Firstly, I would like to thank you for giving us the opportunity to revise. I have been revised according to the reviewer’s opinion and marked it in red. The specific reply are as follows:

Reviewer #2:

Questions 1. The introduction is too short and could be improved.

Answer 1: Thank the reviewer for these kindly suggestions, we have been revised.

Questions 2. I think that Fisher's LSD test was the best choice. This is the most liberal post-hoc test that easily finds significant differences. I recommend to use the more conservative Tukey's test to identify only those significant differences which most likely represent biological meaning.

Answer 2: We have been revised. We selected Duncan’s multiple comparison test according our results.

Reviewer 3 Report

Authors of this work used common biotechnology methods and standard hormone ratios for herbaceous species to increase the number of buds and shoots. Plants belonging to this genus propagate well from leaves even at domestic conditions.

Undoubtedly, the work described in this manuscript is important for the mass propagation of valuable cultivars and genotypes of this genus, but is very far from such genetic approaches as protoplast fusion and genetic transformation. In order to understand how the transformed plants will respond to these hormones, separate studies will be required depending on the technologies used.

What did authors mean by “wild” leaves? Its incorrect. The word (term) must be changed for any journals.

Author Response

Dear editors:

Firstly, I would like to thank you for giving us the opportunity to revise. I have been revised according to the reviewer’s opinion and marked it in red. The specific reply are as follows:

Reviewer #3:

Question 1 Undoubtedly, the work described in this manuscript is important for the mass propagation of valuable cultivars and genotypes of this genus, but is very far from such genetic approaches as protoplast fusion and genetic transformation. In order to understand how the transformed plants will respond to these hormones, separate studies will be required depending on the technologies used.

Answer 1: Thank you very much. We've changed our way of saying.

Question 2. What did authors mean by “field- grown” leaves? Its incorrect. The word (term) must be changed for any journals.

Answer 2: We have been revised. We changed it to “field-grown leaves”

Round 2

Reviewer 1 Report

Some of the suggestions were implemented but some still need application as stated below.

The Authors should consider language checking by a native speaker and removal of editorial mistakes. Experimental design should be clarified in the text.

Introduction

Q1 – changes done in the text (red font) need English polishing.

‘in vitro’ – lacks italics.

Results

Question 2 – Similar explanation should be put in the text of the manuscript and appropriate literature cited. I suggest Materials and Methods section.

Question 3 – The experimental design needs short clarification in the text of the manuscript.

2.1

Question 4. Table 1 – changes done in the text (red font) need English correction.

‘the health degree of the shoots’ – is still unclear and needs explanation in the text.

‘“+” indicates shoot growth condition’ – We can always say that we indicated plant condition. I can imagine that you have meant ‘“+” indicates the worst shoot growth condition’. In my opinion this expression is inaccurate and does not truly captures your intention.

2.2

Question 5 – Poor English, I think that it does not mean what you believe it should.

Question 6 – A short explanation should be put in the text of the manuscript (even in the brackets only).

Question 10. “bud elongation and rooting, the bud proliferation coefficient” are mentioned in the text but I cannot see such parameters in the tables. Where are they?

Answer 10: “bud elongation” and “the bud proliferation coefficient” at table 2, “rooting” at Figure 2I red arrow.

I still cannot accept the duality in the text without appropriate explanation. For me, ‘shoot’ is not the same as ‘bud’. In Tab. 2, I can see ‘Shoot proliferation coefficient’ and in the text above the Tab. 2 ‘bud proliferation coefficient’. The nomenclature should be uniformed ar additional data/explanations added to the text.

Question 11 – Additional explanation should be implemented where experimental design was presented.

2.3

Question 4 – This question was also based on the unclear experimental design, which should be deeply improved.

A few of my previous questions were connected with the experimental design and it is a pity that the Authors did not implemented appropriate changes in the revised version of the text but only explained them to me. The experimental design should be clarified in the text otherwise the potential reader will have serious problems with understanding all those changes done in the experimental part (MS changes, doses of growth stimulators, lengths of cultivation, etc.). Authors should rethink the clear and brief way of presentation and explanation.

3.1

Question 2 – ‘on the other hand’ – start the sentence from the capital letter.

Still explanations of abbreviations need improvement, e.g.,

NAA – not explained when first appeared in the text.

‘auxin NAA’ – NAA is enough here. Explanation that it is auxin should appear when first appeared in the text.

3.3

Question 1. – At the beginning of the second paragraph if you write about your experiment you should underline that you referred to your results, e.g., start the sentence “In our research, …”.

4.1

Question 1. The second paragraph – improve English, please. – It still needs improvement, e.g.  ‘Adjust the pH of all media to 5.6 and autoclaving them for 15 minutes at 121℃’. 

4.2

The text still needs editorial correction. Here are explained the abbreviations which were explained earlier in the revised version of the text – ‘thidiazuron (TDZ), zeatin (ZT), kinetin (Kin), or 6-benzylaminopurine (BA)’.

4.3

Question 1 – first – full name, then abbreviation (in brackets).

Please, check where all explanations should appear in the text.

Unnecessary repetitions

Some basic statistical design was not unified. In 4.2 – ‘Three replicates of 30 leaves were cultured in each treatment.’, 4.3 – ‘Each treatment contained six jars with five explants per jar. The experiment was repeated three times, and 4.4 – ‘Three replicates of thirty shoots were cultured in per  treatment”.

All the information should be mentioned only once in 4.5.

Author Response

Dear editors:

Firstly, I would like to thank you for giving us the opportunity to revise. I have been revised according to the reviewer’s opinion and marked it in red. The specific reply are as follows:

Reviewer #1:

First of all, thank you for your careful, earnest and responsible attitude. Thank you for taking time out of your busy schedule to revise my paper carefully. I have made careful modifications according to your suggestion. 

Introduction

Question 1. changes done in the text (red font) need English polishing

Answer 1: Thanks for your suggestion, we have been revised.

Question 2. ‘in vitro’ – lacks italics.

Answer 2: We have revised the “in vitro” in italics

Results

Question 2 – Similar explanation should be put in the text of the manuscript and appropriate

literature cited. I suggest Materials and Methods section.

Question 3 – The experimental design needs short clarification in the text of the manuscript.

Answer 2-3: Thanks for your suggestion, we have revised the materials and methods.

2.1

Question 4. Table 1 – changes done in the text (red font) need English correction.

‘the health degree of the shoots’ – is still unclear and needs explanation in the text.

‘“+” indicates shoot growth condition’ – We can always say that we indicated plant condition. I

can imagine that you have meant ‘“+” indicates the worst shoot growth condition’. In my

opinion this expression is inaccurate and does not truly captures your intention.

Answer 4: We have made some changes and I wonder if this way is satisfactory to you.

2.2

Question 5. Poor English, I think that it does not mean what you believe it should.

Answer 5: We have modified the statement in the article.

Question 6. A short explanation should be put in the text of the manuscript (even in the brackets only).

Answer 6: Thanks for your suggestion, we have been revised.

Question 10. “bud elongation and rooting, the bud proliferation coefficient” are mentioned in the text but I cannot see such parameters in the tables. Where are they? Answer 10: “bud elongation” and “the bud proliferation coefficient” at table 2, “rooting” at Figure 2I red arrow. I still cannot accept the duality in the text without appropriate explanation. For me, ‘shoot’ is not the same as ‘bud’. In Tab. 2, I can see ‘Shoot proliferation coefficient’ and in the text above the Tab. 2 ‘bud proliferation coefficient’. The nomenclature should be uniformed ar additional data/explanations added to the text.

Answer 10: Thanks for your suggestion, we have been revised according to reviewer’s suggestion. “bud proliferation” has been modified to “shoot proliferation”.

Question 11. Why BA and TDZ in Tab. 2 are in other concentrations than in Tab. 1?

Answer 11: Thanks for your suggestion, we have revised.

Results 2.3

Question 4. Why data from Tab. 3 were achieved after 21 days but in Tab. 4 after 2 and 4 weeks?

Answer 4: Thanks for your suggestion, we have agreed in the article.

3.1

Question 2 – ‘on the other hand’ – start the sentence from the capital letter. Still explanations of abbreviations need improvement, e.g., NAA – not explained when first appeared in the text. ‘auxin NAA’ – NAA is enough here. Explanation that it is auxin should appear when first appeared in the text.

Answer 2: We have removed the word " auxin" according to reviewer’s suggestion. NAA-The first explanation appears in 4.3

3.3

Question 1. – At the beginning of the second paragraph if you write about your experiment you should underline that you referred to your results, e.g., start the sentence “In our research, …”.

Answer 1: Thanks for your suggestion, we have been revised.

4.1

Question 1. The second paragraph – improve English, please. – It still needs improvement, e.g.  ‘Adjust the pH of all media to 5.6 and autoclaving them for 15 minutes at 121℃’. 

Answer 1: Thanks for your suggestion, we have been revised.

4.2

The text still needs editorial correction. Here are explained the abbreviations which were

explained earlier in the revised version of the text – ‘thidiazuron (TDZ), zeatin (ZT), kinetin

(Kin), or 6-benzylaminopurine (BA)’.

Answer : Thanks for your suggestion, we have revised the materials and methods.

4.3

Question 1 – first – full name, then abbreviation (in brackets). Please, check where all explanations should appear in the text.

Answer 1: Thanks for your suggestion, we have been revised.

Question: Unnecessary repetitions.Some basic statistical design was not unified. In 4.2 – ‘Three replicates of 30 leaves were cultured in each treatment.’, 4.3 – ‘Each treatment contained six jars with five explants per jar. The experiment was repeated three times, and 4.4 – ‘Three replicates of thirty shoots were cultured in per treatment”. All the information should be mentioned only once in 4.5.

Answer: Thanks for your suggestion, we have deleted the redundant duplicates.

Reviewer 3 Report

Page 7. Discussion: On the other hand....

Good luck

Author Response

Question: Page 7. Discussion: On the other hand....

Answer: Thanks for your suggestion, we have been revised. I wish you all the best!